# How Dimensionality Affects the Structural Anomaly in a Core-Softened Colloid

Leandro B. Krott [1,†] and José Rafael Bordin [2,*,†]

1    Centro de Ciências, Tecnologias e Saúde, Campus Araranguá, Universidade Federal de Santa Catarina, Rua Pedro João Pereira, 150, Araranguá CEP 88905-120, SC, Brazil

2    Departamento de Física, Instituto de Física e Matemática, Universidade Federal de Pelotas, Caixa Postal 354, Pelotas CEP 96001-970, RS, Brazil

\*    Correspondence: jrbordin@ufpel.edu.br

†    These authors contributed equally to this work.

**Abstract:** The interaction between hard core–soft shell colloids are characterized by having two characteristic distances: one associated with the penetrable, soft corona and another one corresponding to the impenetrable core. Isotropic core-softened potentials with two characteristic length scales have long been applied to understand the properties of such colloids. Those potentials usually show water-like anomalies, and recent findings have indicated the existence of multiple anomalous regions in the 2D limit under compression, while in 3D, only one anomalous region is observed. In this direction, we perform molecular dynamics simulations to unveil the details about the structural behavior in the quasi-2D limit of a core-softened colloid. The fluid was confined between highly repulsive solvophobic walls, and the behavior at distinct wall separations and colloid densities was analyzed. Our results indicated a straight relation between the 2D- or 3D-like behavior and layer separation. We can relate that if the system behaves as independent 2D-layers, it will have a 2D-like behavior. However, for some separations, the layers are connected, with colloids hopping from one layer to another, thus having a 3D-like structural behavior. These findings fill the gap in the depiction of the anomalous behavior from 2D to 3D.

**Keywords:** confined colloids; water-like anomalies; molecular dynamics simulation

## 1. Introduction

There are a large number of models for fluids and colloids, ranging from ab initio and atomistic molecular dynamics simulations to continuum models. In between these extrema, symmetric core-softened (CS) potentials are a class of effective models largely employed to study competitive systems, where the two characteristic length scales in the interaction between the particles compete to dominate the fluid structure [1]. Usually, colloids are made of molecular subunits which form a central packed agglomeration and a less dense, more entropic peripheral area. In this way, colloidal particles show distinct conformations that compete to rule the suspension's behavior [2], being an example of system that can be modeled by isotropic CS potentials [3,4].

Although isotropic, the CS class of potentials shows a competition that come from the existence of the two above mentioned characteristic length scales [5–14] or from softened repulsive potentials [15–19]. As examples, experimental works have shown that the effective interaction in solutions of pure or grafted PEG colloids and Ficoll are well described by two length scales potentials [20–24]. Likewise, computational studies indicate that the same type of effective interactions is able to model polymer-grafted nanoparticles [25,26] or star polymers [27].

Furthermore, water, the most common solvent for colloidal suspensions and the most important material on Earth, also shows competition between two structures originating from the hydrogen bonds breaking and forming between water tetramers [28–30]. As a

consequence, water shows more than 70 anomalous behaviors due the struggle between two conformations trying to dominate the fluid structure [31]. Anomalies are properties that deviate from the expected and observed behaviors of most materials. Certainly, the most well known water anomaly is the density anomaly: normal liquids contract upon cooling at constant pressure, while anomalous fluids expand as the temperature decreases. This anomaly leads to the maximum value in the density of water, 1 g/cm$^3$ at 4 °C and a constant pressure of 1 atm [32]. Another interesting property of anomalous fluids is the increase in the molecular disorder (or increases in entropy) as the density increases—the opposite from the expected, which is that a higher order (or lower entropy) is obtained as the density grows [33]. In this sense, CS potentials are simple and effective models that allow us to unveil the mechanisms behind the behavior of complex fluids as water and hard core–soft shell colloidal suspensions [34–48].

The shape and the type of interaction between colloidal particles combined with the presence of distinct confining surfaces play major roles in self-assembly, phase separation, interfacial activities, and anomalous behaviors of systems modeled by CS potentials [49–53]. Those peculiarities are in agreement with several experiments involving colloidal clusters, the formation of self-assembled mesophases, layering packing, and the synthesis of new structures using amphiphilic molecules [54–59]. Additionally, the structure of colloidal systems has been extensively studied by molecular dynamics and Monte Carlo simulations since hierarchical assembly in a specific patterns at the micro- or macroscopic scale from chemical building blocks can lead to new technologies. In this sense, many distinct of patterns have been observed for 3D bulk [42] and 2D bulk [60–62] systems, with confinement being a promissory approach to controlling the self-assembly and obtaining a specific macrostructure system [52,63–71]. In addition, significant differences are reported when going from a 2D to a quasi-2D system. When confined in very small slit pores, CS potentials can present a 2D-like behavior, but increasing the size of the slit pores affects the solid–liquid transition at low densities due the out-of-plane motion of the particles [71]. Likewise, the quasi-2D limit can affect the temperature of the maximum density observed in CS models [72] and the crystalline structure [73].

Among the many CS models, de Oliveira and co-authors proposed a CS ramp-like potential which have thermodynamic, dynamic, and structural anomalies obeying the same hierarchy observed in water [74,75]. Since then, effects of hydrophobic and hydrophilic confinement through plates and nanotubes on those anomalies and in solid–liquid and surface transitions have also been studied [76–79]. They show that entropic contributions from the flexibility of the confining walls can break the dynamic and structural anomalies of the CS fluid [80], as well induce a superdiffusive to diffusive first-order phase transition [81]. On the other hand, rigid walls can induce surface phase transitions related to specific patterns [82] and establish a new region of structural anomaly [83]. More recently, Bordin and Barbosa [84] studied the 2D case of this CS fluid. They found a second region of density, diffusion, and structural anomaly at high densities. The first region of anomalies is associated with the competition between scales of CS potential, while the second one is generated by a re-entrant fluid marking an order–disorder transition. In addition, an inversion of the hierarchy of these anomalies was observed, in agreement with results from Dudalov et. al [85]. Later, Cardoso and co-workers [62] found a third region of structural anomaly at higher densities. They showed that each anomalous behavior can be associated to order–disorder transitions and re-entrant fluid phases that separate two solid crystalline regions and to distinct aggregate sizes in the fluid phase.

But where is the boundary between the 3D behavior, with one region of structural anomaly in the fluid phase associated with a crystalline–amorphous solid transition [75,86] and the 2D behavior, where up to three anomalous structural regions were found [49,62]? To unveil the location of this boundary, we expand our previous works [64,82,83] to see what happens in the quasi-2D limit when the CS model is confined between two flat, highly repulsive solvophobic walls. Our findings indicate that the limit between 2D-like or 3D-like

behavior is associated with the layering and with the existence of order–disorder transitions in the layers.

This paper is organized as follows: in Section 2, we present the model; in Section 3, we present the methods and simulation details; in Section 4, we discuss the results; and in Section 5, we present the conclusions.

## 2. The Model

The colloidal system is modeled by spherical particles confined between smooth parallel plates. The particle–particle (P-P) interaction occurs through a two length scales potential, given by

$$\frac{U(r_{ij})}{\varepsilon} = 4\left[\left(\frac{\sigma}{r_{ij}}\right)^{12} - \left(\frac{\sigma}{r_{ij}}\right)^{6}\right] + a\exp\left[-\frac{1}{c^2}\left(\frac{r_{ij} - r_0}{\sigma}\right)^2\right] \qquad (1)$$

where $r_{ij} = |\vec{r}_i - \vec{r}_j|$ is the distance between two particles, $i$ and $j$. The first term is a standard Lennard-Jones (LJ) 12-6 potential, where $\varepsilon$ is the depth and $\sigma$ is the particle's diameter. The second one is a Gaussian centered on radius $r_0$ and width $c\sigma$. In this work, we obtain the ramp-like shape indicated by the blue line in Figure 1 with the following parameters: $a = 5$, $r_0/\sigma = 0.7$, and $c = 1$. This specific interaction potential reproduces water-like anomalies, such as density, diffusion, and translational and orientational anomalies [74,75]. The P-P interaction potential has a cutoff of $r_c/\sigma = 3.5$.

The wall–particle (W-P) interaction is given by a strong repulsive potential, the so-called R6 potential [79,83]:

$$U_{R6}(z) = \begin{cases} 4\varepsilon(\sigma/z)^6 + 0.1875\varepsilon(z/\sigma) - U_{R6c}, & z \leq z_c \\ 0, & z > z_c \end{cases} \qquad (2)$$

where $z$ is the distance between the particles and the walls, $z_c = 2.0\sigma$, and $U_{R6c} = 4\varepsilon(\sigma/z_c)^6 + 0.1875\varepsilon(z_c/\sigma)$. Figure 1 shows the profile of P-P and W-P interaction potentials. All quantities used in this work are given in LJ units [87], for example, distance $r^* = r/\sigma$ and temperature $T^* = k_B T/\varepsilon$. The symbol * will be omitted for simplicity.

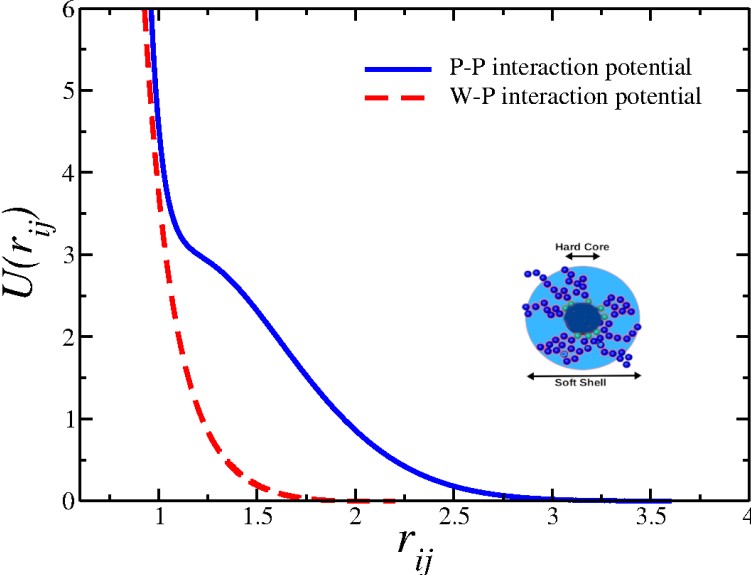

**Figure 1.** Particle–particle (P-P) interaction potential (solid blue line) and wall–particle (W-P) interaction potential (dashed red line). Insert shows the schematic depiction of a hard core–soft shell polymer-coated nanoparticle as a core-softened effective colloid [25].

### 3. Methods and Simulation Details

We perform molecular dynamics simulations in the $NVT$ ensemble. A total of 2028 particles were confined inside pores built with two parallel, smooth, fixed plates. The walls were held fixed in the $z$ direction and separated by a distance $L_z$. $L_z$ varied from 4.8 to 6.2. Those specific separations were chosen so we could study the system with two or three particles layers. For each fixed $L_z$, we changed the simulation box size, given by $L$, in order to simulate different total densities. Total densities are defined as $\rho = N/(L_{ze}L^2)$, where $L_{ze} = L_z - \sigma$ is the effective distance between the plates. Periodic boundary conditions were used in the $x$ and $y$ directions in order to simulate infinite slab systems. Our simulations were performed along the isotherm $T = 0.150$, and we analyzed the effects of the increase in density $\rho$ and the decrease in wall separation $L_z$.

Briefly stated, molecular dynamic simulations consist of solving Newton's equations of movement for $N$ particles (whose initial positions and velocities are known) interacting through a known force field. Positions and velocities were calculated for every time step and stored after the system reached its equilibrium state. In this work, we analyzed the total energy, kinetic energy, and pressure as functions of time to ensure the equilibration. Instantaneous temperature was obtained by the equipartition theorem of energy, which relates temperature to the degree of freedom for translational motion and kinetic energy. In relation to pressure, it is possible to define two types: parallel pressure, which corresponds to the evaluation of the Virial teorem for $x$ and $y$ directions, and perpendicular pressure, which uses the ratio between the force exerted by the walls and its area.

The equations of motion were integrated through the Velocity Verlet algorithm, and the Nose–Hoover heat-bath with a coupling parameter Q = 2 was used to keep $T$ fixed. Equilibration was reached after $2 \times 10^6$ time steps of simulation and followed by $1 \times 10^6$ time steps for the production of results, with a increment of time $\delta t = 0.001$ at each time step.

The system structure was analyzed by the transversal density profile and the lateral radial distribution function (LRDF), in addition to some correlated quantities, such as the translational order parameter, the two-body entropy, and the cumulative two-body entropy. The lateral radial distribution function $g_\parallel(r_\parallel)$ is defined as

$$g_\parallel(r_\parallel) \equiv \frac{1}{\rho^2 V} \sum_{i \neq j} \delta(r_\parallel - r_{ij}) \left[ \theta\left( \delta z - |z_i - z_j| \right) \right] \tag{3}$$

where $r_\parallel$ is the parallel distance between particles in the $x$ and $y$ directions. Considering that systems structure themselves in layers, the Heaviside function, $\theta(x)$, restricts the sum of particle pairs in the same slab of thickness $\delta z/\sigma = 1.0$ for contact and central layers.

In order to describe the connection between structure and thermodynamics, the LRDF was subsequently used to compute the excess entropy, $s_{ex}$. It is obtained by counting all accessible configurations for a fluid and comparing it with the ideal gas entropy [88]. Therefore, the excess entropy is a negative quantity once real liquids become more ordered than the ideal gas. It is relevant to notice that $s_{ex}$ grows under heating, as the full entropy $S$ does, and $s_{ex} \to 0$ as temperature goes to infinity at a fixed pressure or density—in this extremum, the system approaches an ideal gas [89,90]. The excess entropy can be obtained exactly if the equation of state is known [91]. Another approach consists of the systematic expansion of $s_{ex}$ in terms of two-particle, three-particle contributions, etc.:

$$s_{ex} = s_2 + s_3 + s_4 + ... \tag{4}$$

The two-body entropy, the dominant contribution to excess entropy [92,93], can be obtained from the LRDF using [93]

$$s_2 = -\frac{\rho_l}{2} \int \left[ g_\parallel(r_\parallel) \ln(g_\parallel(r_\parallel)) - g_\parallel(r_\parallel) + 1 \right] d^2 r_\parallel. \tag{5}$$

We should address that $s_2$ is not a thermodynamic property of the system, but it is a powerful tool to analyze structural characteristics of the core-softened system. The normal behavior of $s_2$ is to decrease with density or pressure at constant temperature. However, in anomalous systems, $s_2$ increases with the density or pressure at some region of its phase diagrams—the anomalous region. Like the translational order parameter, $s_2$ also presents a double region of anomaly for confined systems [83] and a triple anomalous region for the 2D case [62].

To analyze the long-range translational ordering, we evaluated the cumulative two-body entropy, defined as [94]

$$C_{s2}(R) = -\pi \int_0^R \left[ g_\parallel(r_\parallel) \ln(g_\parallel(r_\parallel)) - g_\parallel(r_\parallel) + 1 \right] r_\parallel dr_\parallel, \tag{6}$$

where $R$ is the upper integration limit. This quantity give us information about the long-range translational order. For fluids and amorphous phases that do not present long-range information, $C_{s2}(R)$ converges, while for solids that present long-range information, $C_{s2}(R)$ diverges.

Another quantity obtained from the LRDF is the translational order parameter, defined as [33,95,96]

$$\tau = \int \mid g(\epsilon) - 1 \mid d\epsilon, \tag{7}$$

where $\epsilon = r_\parallel \rho_l^{1/2}$. We calculate the average number of particles per layer, $< N_l >$, and define the layer density as $\rho_l = < N_l > / (L^2 \delta z)$.

The translational order parameter provides us information about the structure of the system. Gas and liquids have low values of $\tau$, while crystal and amorphous solids present high values of $\tau$. In normal fluids, $\tau$ increases monotonically with density or pressure at constant temperature, but in anomalous fluids, $\tau$ decreases with density or pressure for some temperatures. The current potential presents one region of anomaly in the 3D bulk system [75] due the competition between scales. For confined systems and considering the particles near the walls (contact layers), in addition to the anomalous region produced by the competition between scales, another one appears, induced by nanoconfinement at low densities [83]. More recent works [62,84] showed that dimensionality leads to extra structural anomalies at higher densities and low temperatures.

## 4. Results

We can imagine the 2D limit as a single-layer limit. In this sense, we have investigated systems with two, two-to-three, or three layers of confined CS colloids, depicted by their transversal density profiles in Figure 2a. Layers near the walls are called contact layers, while the others, if they exist, are called central layers. Systems with two well-defined layers are observed for the most densities explored for plates separated by $L_z = 4.8$ and 5.0, while three well-defined layers occur for $L_z = 5.8$ to 6.2. Intermediate values of $L_z$ lean over to present a transition between two-to-three layers for most densities simulated. In Figure 2b,c we show examples of snapshots of systems with two ($L_z = 4.8$) and three ($L_z = 6.0$) well-defined layers, respectively.

Our first analysis is related to particles located near the walls, namely, the contact layers. As already mentioned, this model has one region of anomaly in a translational order parameter, $\tau$, for a 3D bulk system [75,86] due to the competition between scales and three regions of anomaly for 2D bulk system [84], one of them at lower density due the competition between scales. The origin of the other two anomalous regions, at intermediary ($\rho \approx 0.42$) and high ($\rho \approx 0.65$) densities, are related to the order–disorder transition and the re-entrant fluid phases [62]. In a previous work, we have shown that confining fixed walls also induce a new region of anomaly at low densities, $\rho_l < 0.350$ [83], that was not observed in the 2D limit. Now, we present the behavior of $\tau$ for higher densities that have been studied in the past, and we found the peculiar profile presented in Figure 3a. The

figure shows the parameter $\tau$ for the contact layer as a function of the layer density, $\rho_l$, for several separations of plates at $T = 0.150$. Systems with plates separated by $L_z = 5.2$, 5.3, 5.5, and 5.6, drafted in orange lines, do not have anomalies for $\rho_l \approx 0.400$, that is, they present 3D-like fluid (or disordered) behavior. On the other hand, the other curves corresponding to $L_z = 4.8, 5.0, 5.8, 6.0,$ and 6.2 present a third region of anomaly, which means a 2D-like (ordered) behavior for this gap of densities ($0.400 \leq \rho_l \leq 0.500$), including the same values of $\rho_l$ compared to those found for 2D systems [62,84].

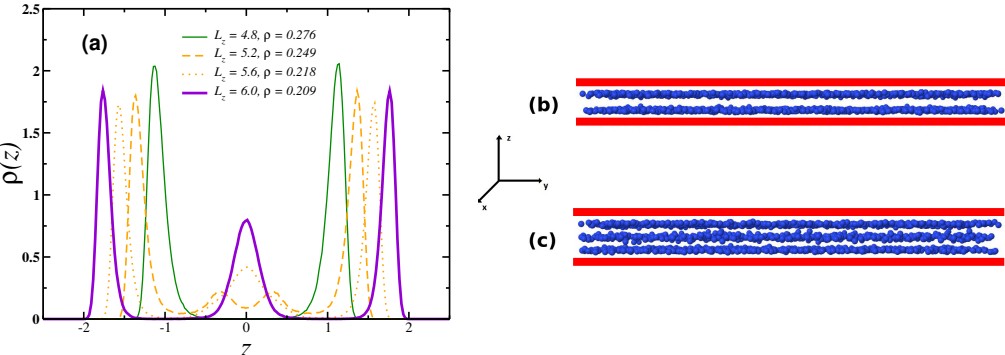

**Figure 2.** (**a**) Transversal density profile for some systems that present the formation of two (green solid line), two-to-three (orange dashed and dotted lines), and three layers (violet solid line) of particles between plates. Examples of snapshots of systems with (**b**) two layers for plates separated by $L_z = 4.8$ and (**c**) three layers for plates separated by $L_z = 6.0$. The fluid particles are in blue and the confining walls are in red.

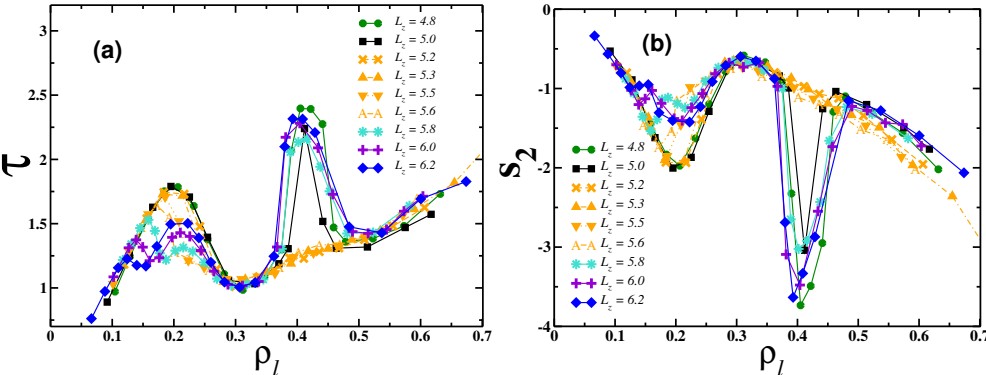

**Figure 3.** (**a**) Translational order parameter $\tau$ and (**b**) two-body entropy $s_2$ as functions of layer density $\rho_l$ at different separations of plates ($L_z$) and $T = 0.150$. Both quantities are related to contact layers.

Another quantity analyzed was the two-body entropy, which gives us structural information, because its definition is based on the radial distribution function (Equation (5)). Normal fluids lean over to present values of $s_2$ that monotonically decrease with density. $s_2$ also presents anomalous behavior in 3D and 2D bulk and 3D confined systems. Two regions of anomaly in $s_2$ were already shown in the past for $\rho_l$ until $\approx 0.350$. Now, going to higher layer densities, we found the profile seen in Figure 3b. In the current work, we explored higher densities and found 2D-like and 3D-like behaviors, depending on the $L_z$ studied. As observed for the translational order parameter, $s_2$ also presented a 2D-like behavior at high densities ($0.400 \leq \rho_l \leq 0.500$) for $L_z = 4.8, 5.0, 5.8, 6.0,$ and 6.2 and a 3D-like behavior for $L_z = 5.2, 5.3, 5.5,$ and 5.6.

To understand why some cases have one or more structural anomalous regions, as well as their origins, we analyze the transversal density profile ($\rho(z)$ as a function of $z$), the lateral radial distribution function ($g(r_{||})$ as a function of $r_{||}$), and the cumulative two-body entropy ($|C_2(r_{||})|$ as a function of $r_{||}$) for three situations: the lowest separation $L_z = 4.8$, corresponding to a system that present two layers; the separation $L_z = 6.0$, which present three layers; and an intermediate case, $L_z = 5.5$, that has a two-to-three layers transition.

Let us begin by analyzing the data for $L_z = 4.8$ (Figure 4). The solid green lines correspond to densities in the region of highly ordered particles, $0.390 \leq \rho_l \leq 0.441$. The dashed light green lines correspond to some densities above the maximum in $\tau$ and minimum in $s_2$, $0.459 \leq \rho_l \leq 0.523$, while the dashed-dotted light green lines correspond to some densities below the maximum in $\tau$ and minimum in $s_2$, $0.282 \leq \rho_l \leq 0.375$. In Figure 4a, the transversal density profiles show two well-defined layers, evidenced by the zoom-in given in the inset graph, which suggests ordered particles. Figure 4b shows the lateral radial distribution function and makes evident the ordering of particles in the third maximum of $\tau$ (solid green lines) compared to vicinity densities (disorder–order–disorder transition), the same behavior observed in 2D systems [62,84]. The long-range translational behavior, given by the cumulative two-body entropy, $|C_2(r_{||})|$, confirms the tendency of solidification at densities of $0.390 \leq \rho_l \leq 0.441$.

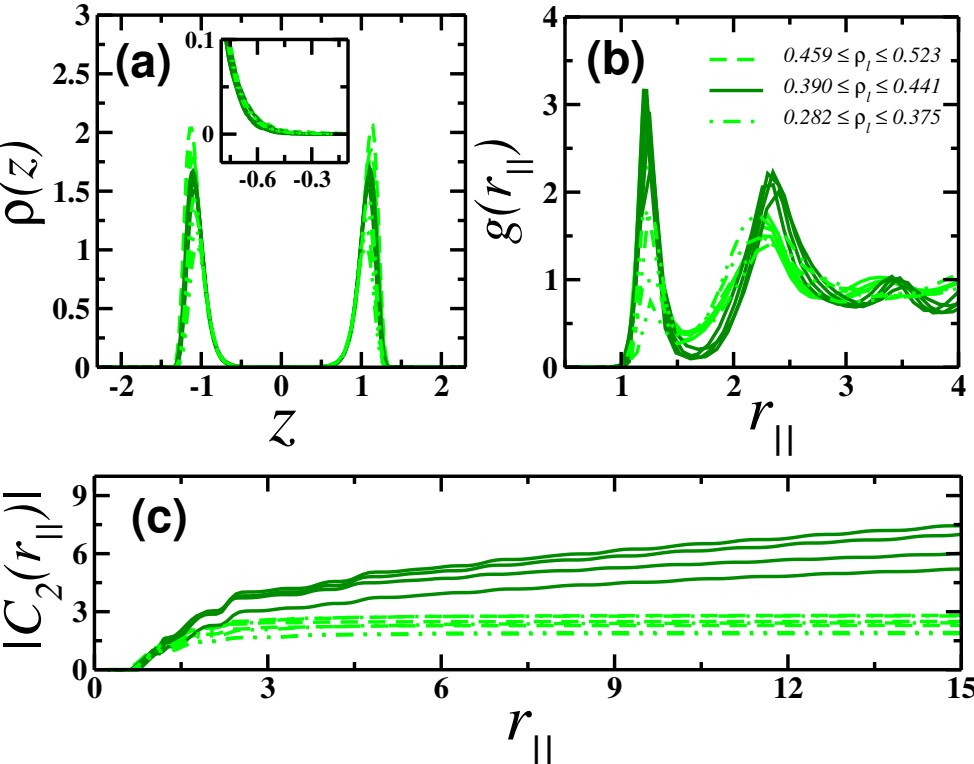

**Figure 4.** Quantities related to $L_z = 4.8$ at $T = 0.150$: (**a**) transversal density profile, (**b**) lateral radial distribution function, and (**c**) cumulative two-body entropy.

The same behavior was found for another system that presented 2D-like behavior at high densities, $L_z = 6.0$, shown in Figure 5. The solid violet lines correspond to densities in the region of ordered particles (third maximum in $\tau$ and minimum in $s_2$), $0.381 \leq \rho_l \leq 0.458$. The dashed magenta lines correspond to some densities above the maximum in $\tau$ and minimum in $s_2$, $0.490 \leq \rho_l \leq 0.545$, while the dashed-dotted magenta lines correspond to some densities below the maximum in $\tau$ and minimum in $s_2$, $0.265 \leq \rho_l \leq 0.367$. Despite the fact that the colloids here are arranged in three well-defined layers, as seen in Figure 5a and its inset graph, the basic mechanism behind the structural anomalous region is the same as the one found in the previous case, Figure 4. Both $g(r_{||})$ and $|C_2(r_{||})|$ indicate a order–disorder transition in the contact layers, as we can see in Figure 5b,c, respectively. Therefore, the origin is the same as the one observed in the 2D system: an ordered–disordered transition leads to the water-like structural anomaly.

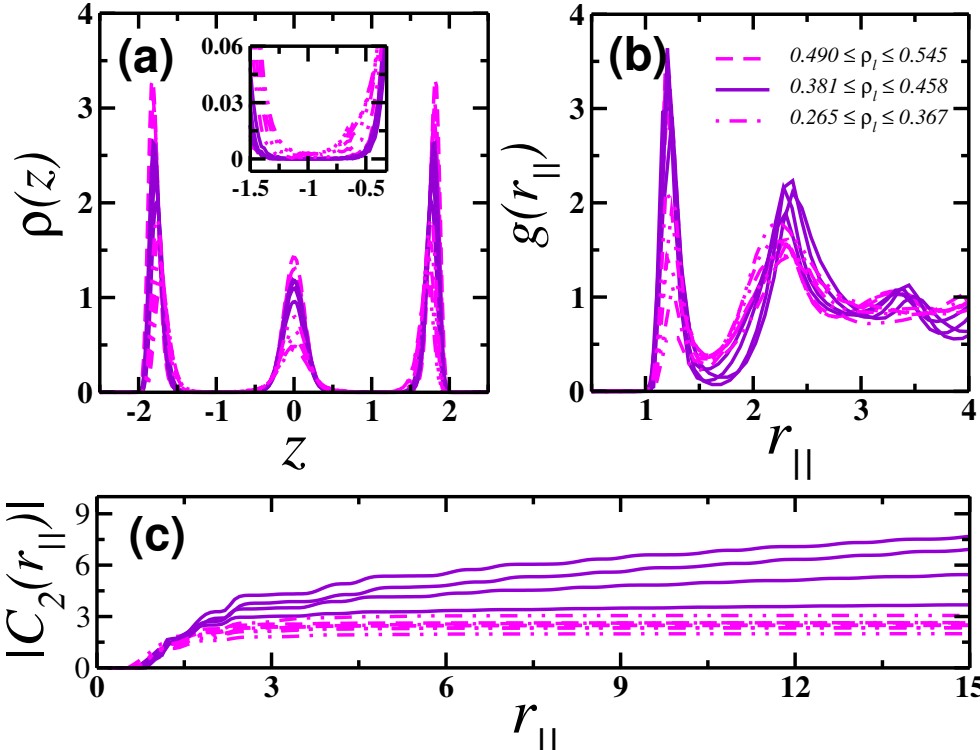

**Figure 5.** Quantities related to $L_z = 6.0$ at $T = 0.150$: (**a**) transversal density profile, (**b**) lateral radial distribution function, and (**c**) cumulative two-body entropy.

So far, we have observed a 2D-like behavior and mechanism for the structural anomalies in systems with two and three layers. But what happens in systems with a single anomalous region? To clarify this point, we show in Figure 6 the $L_z = 5.5$ case. The solid orange lines correspond to densities in the region that should present ordered particles (third maximum in $\tau$ and minimum in $s_2$ for another cases), $0.394 \leq \rho_l \leq 0.465$. The dashed brown lines correspond to some densities above $0.512 \leq \rho_l \leq 0.571$, while the dashed-dotted brown lines correspond to some densities below the region where the ordering of particles was supposed to happen, $0.249 \leq \rho_l \leq 0.364$. In Figure 6a, the transversal density profiles show a two-to-three layers transition. However, as indicated by the zoom-in in the inset graph, the density does not go to zero in the region between the layers. Unlike the previous cases (Figures 4 and 5), here, there is a connection between the contact and central layers. This indicates that some particles from the central layer jump to the contact layer, and the other way around as well. This is a consequence of the frustration added by this specific separation length that forces the colloids to organize themselves in such way that the separation between two layers is smaller than the second length scale in the CS potential but bigger than the first length scale. This frustration prevents the system from relaxing to an organized phase. This is also clear in Figure 6b. The lateral radial distribution function does not go to zero for separations between the first and second peaks, preventing the ordered phase. This is corroborated by the cumulative two-body entropy, $|C_2(r_{||})|$, that presents a liquid (disordered) profile for all densities, as shown in Figure 6c. Without an order–disorder transition, the system leans to present a 3D-like behavior.

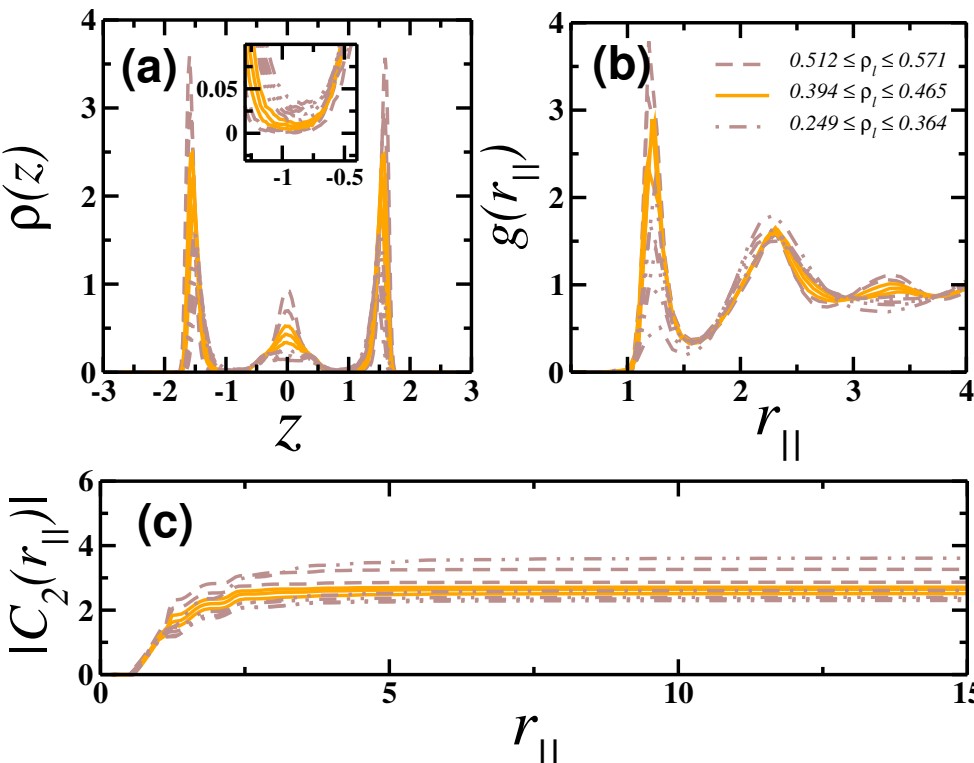

**Figure 6.** Quantities related to $L_z = 5.5$ at $T = 0.150$: (**a**) transversal density profile, (**b**) lateral radial distribution function, and (**c**) cumulative two-body entropy.

The central layers also were analyzed for the separations from $L_z = 5.5$ to 6.2. Pores with a separation of $L_z < 5.5$ have two layers, or a transition between two-to-three layers. In Figure 7a,b, we show both the translational order parameter and two-body entropy as a function of the central layer's density, respectively. As already observed for contact layers, systems at $L_z = 5.8$, 6.0, and 6.2 that present order–disorder transition have anomalous behavior around $\rho_l = 0.4$, while systems at $L_z = 5.5$ and 5.6, presenting frustration due the hopping of particles between layers, does not show anomalous behavior. The results for the central layers are evidence that 2D-like behavior at high densities is connected to the movement in the confined direction.

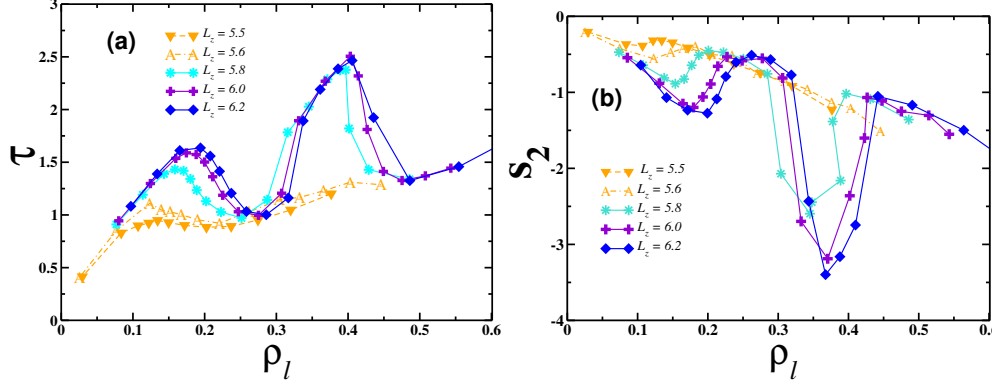

**Figure 7.** (**a**) Translational order parameter and (**b**) two-body entropy as function of central layer density. The central layers were analyzed at $T = 0.150$.

## 5. Conclusions

Core-softened colloids can have unique and peculiar structural properties. The model employed in this study shows multiple regions of structural anomalous behavior in the 2D system, while the 3D case has only one. In this paper, we seek to unveil the details

about the structure of a CS model in the quasi-2D limit. In this sense, we have studied systems confined in nanopores modeled by smooth parallel walls separated by different distances $L_z$. The separations were chosen so that the particles can be arranged in layers of two, two-to-three, or three particles, which correspond to $L_z$ values between 4.8 and 6.2. The interaction between particles and walls is given by the solvophobic, purely repulsive R6 potential.

Our results have shown that all contact layers show low-density structural anomalies related to the competition between the characteristic interaction length scales. However, the anomalies due the order–disorder transition at higher densities, as observed in the 2D system, have not been observed for all cases. Specifically, inside pores with walls separation from $L_z = 5.2$ to 5.6, the particles hop from one layer to another. This hopping from contact layers to central layers frustrates the order–disorder transition, preventing the emergence of an additional region of structural anomalies, approaching the 3D limit. Without this hopping of particles, the layers are disconnected and behave like independent 2D systems, allowing the order–disorder transition and its consequent anomalies. In addition, when the particles are arranged in three layers, the behavior of the central layer is not directly influenced by the walls, and even that follows the same 2D-like behavior observed for contact layers at high densities. It is an evidence that walls do not rule the anomalies at high densities, but the order–disorder transition observed in those particular model does. As consequence of this, the double anomaly observed in the contact layers at low densities due the two-to-three layer transition is absent in the central layers.

**Author Contributions:** L.B.K.: Conceptualization, Methodology, Software, Data curation, Validation, Formal analysis, Investigation, Writing—original draft. J.R.B.: Conceptualization, Methodology, Software, Investigation, Resources, Writing—review & editing, Visualization, Supervision, Project administration, Funding acquisition. All authors have read and agreed to the published version of the manuscript.

**Funding:** Without public funding, this research would be impossible. JRB is grateful to the CNPq, proc. 304958/2022-0, and FAPERGS, TO 21/2551-0002024-5, for the funding support.

**Data Availability Statement:** All data are available upon reasonable request.

**Conflicts of Interest:** The authors declare no conflict of interest.

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
