# Peer review of "How Dimensionality Affects the Structural Anomaly in a Core-Softened Colloid"

_colloids, doi:10.3390/colloids7020033_

Round 1

Reviewer 1 Report

In this article, the authors study the behavior of a system with a core-softened potential enclosed in a slit pore formed by rigid walls. The particles of the system interact with the walls by means of the repulsive potential R6 introduced in the previous works of the authors. The considered potential was comprehensively studied in the past works of the authors in the case of two and three dimensions. Systems with this potential demonstrate waterlike anomalies, and earlier the authors showed that the anomalous behavior depends on the dimension of the system and differs for the cases of 2D and 3D dimensions. Similar behavior was also previously found in the works of other authors for a similar system.

In the article under consideration, the authors analyze in detail the dependence of the behavior of a structural anomaly on the number of layers formed depending on the distance between the walls. As the pore width increases, the number of layers changes from two to three, and it was found that there is a transition range of width values in which the third layer appears. In this range the second region of the structural anomaly disappears, which is characteristic of quasi-two-dimensional systems with exactly two and three layers. The authors offer a qualitative explanation of this behavior in terms of particle hopping between layers.

The result obtained is quite interesting and deserves publication in Colloids Interfaces. The article is written clearly and understandably, and can be published in its present form.

Author Response

We thanks the referee for the critical review of our work.

Reviewer 2 Report

The authors present a study on the structural organisation of colloids with a hard core and a soft shell under confinement in a narrow slit. Depending on the height of the slid, the particles fit nicely and from 2 or 3 layers of 2D fluid, while at intermediate heights they fit poorly and are more like a 3D fluid. The observations are nice, but what does the reader learn that has not already been explained in previous work, e.g. the earlier study [83] by the same group? Does the additional behaviour reported in this manuscript, i.e. an extra peak at a density of about 0.4, even depend on the particular potential chosen, as simple Lennard-Jones particles will also display transitions between good and poor stacking between two walls?

Minor comments:
* The Gaussian in Eq 1 does not have a width of c, as reported in line 92, but of c * sigma.
* The definition in Eq 3 is akward, as theta( | z_i - z_j | ) is an involved way of writing 1. To delimit the analysis to particles with closely matching z coordinates, simply replace the entire square-bracketed factor with theta( delta z - | z_i - z_j | ).
* What are the actual "intermediate" (no -ly) densities reported on in Figs 4, 5 and 6?
* Fig 4 does not show rho x z (line 184) but rho against z.
* Yellow is a poor colour for plots.

The manuscript would benefit from careful proof-reading by a native speaker.

Author Response

We thanks the referee for the critical review of our work. In the attached file reviewer will find a detailed reply to all the comments and criticisms, as well
eventual changes in our manuscript.
